# Attending on Multilevel Structure of Proteins enables Accurate Prediction of Cold-Start Drug-Target Interactions

## Abstract

Cold-start drug-target interaction (DTI) prediction focuses on interaction between novel drugs and proteins. Previous methods typically learn transferable interaction patterns between structures of drug and proteins to tackle it. However, insight from proteomics suggest that protein have multi-level structures and they all influence the DTI. Existing works usually represent protein with only primary structures, limiting their ability to capture interactions involving higher-level structures. Inspired by this insight, we propose ColdDTI, a framework attending on protein multi-level structure for cold-start DTI prediction. We employ hierarchical attention mechanism to mine interaction between multi-level protein structures (from primary to quaternary) and drug structures at both local and global granularities. Then, we leverage mined interactions to fuse structure representations of different levels for final prediction. Our design captures biologically transferable priors, avoiding the risk of overfitting caused by excessive reliance on representation learning. Experiments on benchmark datasets demonstrate that ColdDTI consistently outperforms previous methods in cold-start settings.

## 1 Introduction

Identifying drug–target interactions (DTIs) is fundamental to drug discovery, yet traditional wet-lab experiments are costly and time-consuming, which often spanning years or decades (Hughes et al., 2011). Recently, in silico DTI prediction has greatly improved efficiency by prioritizing candidate interactions, allowing wet-lab studies to focus on a smaller and more promising subset of drug–target pairs. However, an urgent challenge in practice is cold-start DTI prediction, which refers to inferring interactions involving newly discovered drugs or newly identified target proteins. This requires the computational models have the ability to generalize beyond the observed interactions and make reliable predictions. Existing methods typically learn interaction patterns from known pairs and transfer them to unseen ones, but their generalization ability remains limited in cold-start scenarios.

Traditional DTI prediction methods fall into graph-based and structure-based categories. Graph-based models struggle in cold-start scenarios due to a lack of informative neighbors for new nodes. Therefore, existing efforts to address cold-start DTI mainly focus on structure-based approaches, which exploit the intrinsic features of drugs and proteins. GraphDTA (Nguyen et al., 2020) uses GNNs for drug molecular graphs and CNNs for protein sequences to predict binding affinity. MolTrans (Huang et al., 2021) utilizes large unlabeled data to identify key substructures and compute interactions. TransformerCPI (Chen et al., 2020) encodes proteins with word2vec embeddings and drugs with GCN-based atomic features, and employs a modified Transformer encoder–decoder to model compound–protein interactions. HyperAttentionDTI (Zhao et al., 2022) embeds atoms and residues, extracts fragment features with stacked 1D CNNs, and applies a hyper-attention mechanism for fine-grained interactions between drug fragments and protein subsequences. Drug-BAN (Bai et al., 2023) employs bilinear attention to capture interactions between drug substructures and protein subsequences More recently, MlanDTI (Xie et al., 2024) builds on pretrained encoders and fuses features from different network layers, avoiding models over-rely on drug patterns and neglect protein information. Despite these advances, most structure-based methods remain restricted to shallow representations and overlook the hierarchical organization of proteins, which limits their ability to generalize in cold-start scenarios.

These limitations highlight the need for models that go beyond sequence embeddings and incorporate biologically grounded structural priors, capturing interactions across different protein levels. As illustrated in Figure 1, proteins exhibit a natural hierarchy of structural levels and drug–protein interactions can occur at different levels of granularity. Building upon these biological and physical insights, we develop our method by explicitly incorporating such structural priors, aiming to provide a more generalizable solution to the cold-start DTI prediction problem.

In this paper, we introduce ColdDTI, a novel framework designed for cold-start DTI prediction. Inspired by MlanDTI, we also used pre-trained models to perform embeddings on drugs and proteins. For drugs, ColdDTI represents molecules at two granularities: token-level embeddings derived from Simplified Molecular Input Line Entry System (SMILES) sequences and holistic molecular representations. For proteins, it explicitly considers hierarchical biological structures, including primary sequences, secondary motifs, tertiary substructures, and quaternary global embeddings. To bridge these modalities, ColdDTI constructs cross-level interaction attention maps that align drug representa-

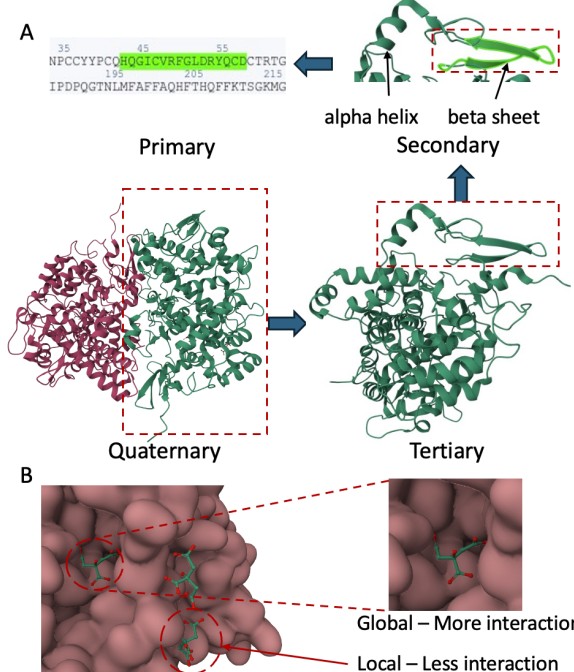

Figure 1: A real example of the Human COX-1 Crystal Structure with drug FLC (A) Protein exhibits hierarchical structures from primary to quaternary levels. (B)Drug–protein interactions vary in granularity, with global interactions involving broader binding surfaces and local interactions reflecting weaker contacts.

tions at both fragment and global levels with protein structures across multiple hierarchical scales, capturing complementary relationships that single-level models tend to ignore. Furthermore, it employs an adaptive fusion mechanism that dynamically balances contributions from different drug granularities and protein structural levels, improving generalization under cold-start scenarios by flexibly shifting between detailed and holistic perspectives. By jointly leveraging hierarchical protein modeling, cross-level interaction learning, and adaptive fusion, ColdDTI provides a principled solution for predicting interactions in cold start settings, where training data are sparse or entirely missing. Experiments show that ColdDTI achieves superior or comparable performance to the state-of-the-art baselines on key metrics such as AUC across four benchmark datasets, highlighting its strong generalization ability.

The contributions of this paper are as follows:

- We introduce a new paradigm for cold-start DTI prediction by explicitly considering multi-level structure of proteins to improve generalization in scenarios involving novel drugs and proteins.
- ColdDTI extract multi-level structure representations for protein(from primary to quaternary structures) and drugs (local functional groups and global topology). Through hierarchical attention and adaptive fusion, ColdDTI mines interaction patterns involving different-level structures and leverage the mined interactions to fuse multi-level structure representations for final prediction.
- Extensive experiments on multiple benchmark datasets demonstrate that ColdDTI consistently outperforms state-of-the-art methods under various cold-start settings, highlighting its effectiveness in capturing complex hierarchical interaction patterns.

## 2 RELATED WORK

Mainstream approaches for DTI prediction can be broadly divided into two categories. Except for the structure-based methods discussed in Section 1, which focus on modeling the drugs and proteins

structure representations and learning interaction patterns between these structures for binary classification, another dominant paradigm is graph-based methods. Graph-based methods formulate DTI prediction as a link prediction task on heterogeneous networks, where drugs and proteins are represented as nodes. Drug–protein edges are directly constructed from known interaction data in the dataset, while drug–drug and protein–protein edges are often derived from similarity measures. In addition, some studies further incorporate auxiliary biomedical information such as side effects and diseases to enrich the network structure. By propagating information through the network topology, graph-based models aim to infer interactions from observed connectivity patterns.

Existing works typically treat DTI as a link-prediction problem. DTINet (Luo et al., 2017) learns low-dimensional embeddings from heterogeneous networks via random walk and diffusion analysis for DTI prediction. NeoDTI (Wan et al., 2019) integrates multiple relations into a network, learning topology-preserving embeddings. IMCHGAN (Li et al., 2021) models DTIs as multi-channel graphs with graph attention and adversarial learning. SGCL-DTI (Li et al., 2022) enhances embeddings through self-supervised graph contrastive learning on drug–protein bipartite graphs. iGRLDTI (Zhao et al., 2023) mitigates GNN oversmoothing to improve representation learning. GSRF-DTI (Zhu et al., 2024) uses GraphSAGE with random forests for interaction prediction. NASNet-DTI (Zhong & Du, 2025) introduces a node-adaptive depth mechanism to handle oversmoothing and exploit multiple relations. Although graph-based models effectively exploit network connectivity and auxiliary biomedical information, their heavy reliance on existing edges makes them vulnerable in cold-start scenarios, where the sparsity of DTI datasets leaves new drugs or proteins without neighbors. This limitation underscores the need to move beyond relation-driven approaches and directly leverage the intrinsic properties of molecules.

Prior studies demonstrated that specific sub-structures in both drugs and proteins often serve as the key determinants of drug–target interactions (Schenone et al., 2013). Inspired by this insight, many subsequent works have followed this direction. Examples are MolTrans (Huang et al., 2021), TransformerCPI (Chen et al., 2020), HyperAttentionDTI (Zhao et al., 2022) and DrugBAN (Bai et al., 2023). However, many of these models treat drugs and proteins as flat sequences (primary-level), thereby ignoring their structural hierarchies. Some approaches, such as GraphDTA (Nguyen et al., 2020), represent drugs as molecular graphs and preserve their 2D topological information, yet this remains insufficient for capturing the complexity of interactions. MlanDTI (Xie et al., 2024) aligned representations across different layers in deep neural network, which improves performance but still lacks biological interpretability, since its "multi-levels" correspond to network depth rather than structural hierarchies. More recently, EviDTI (Zhao et al., 2025) used both 2D and 3D information of the drugs but still overlooks the informative protein multi-level information. This gap highlights the need for methods that explicitly incorporate biologically meaningful multi-level structures.

# 3 PROBLEM FORMULATION

DTI prediction aims to determine whether a given drug interacts with a specific protein. Following previous works (Luo et al., 2017; Zhao et al., 2022; Bai et al., 2023), we model DTI prediction as a binary classification task, where the interaction between a drug-target pair $(D, T)$ is represented by a label $y$. Specifically, $y = 1$ indicates that an interaction exists between drug $D$ and protein $T$, while $y = 0$ indicates no interaction.

Given a drug $D$, referring previous works (Zhao et al., 2022; Xie et al., 2024), we represent it with SMILES, as a sequence of non-overlapping chemical local structures $D = (s_1, s_2, ..., s_n)$, where $s_j$ corresponds to a local structure (e.g., atoms like C, O, ions such as [NH4+], or atom groups like [NH2]) (Schwaller et al., 2018). Here, $n$ is the number of local structures in $D$. Given a protein $T$, Prior works usually represent it as a sequence of amino acid residues (i.e. primary structure) $T = (a_1, a_2, ..., a_{m_i})$, where $a_j$ is an amino acid residue, and $m$ is the length of $T$. However, $T$ has multi-level structure as introduced in Section 1. To model such multi-level structure, we represent each secondary structure by its starting and end position on residue sequence, as well as its type (e.g. $\alpha$-helix or $\beta$-sheet). Tertiary structure is also represented by its starting and end position. Quaternary structure is in fact the whole protein, thus dose not need extra representation.

ColdDTI addresses the cold-start DTI task, aiming to learn interaction patterns from known drug–target pairs and generalize them to novel pairs involving previously unseen drugs or proteins.

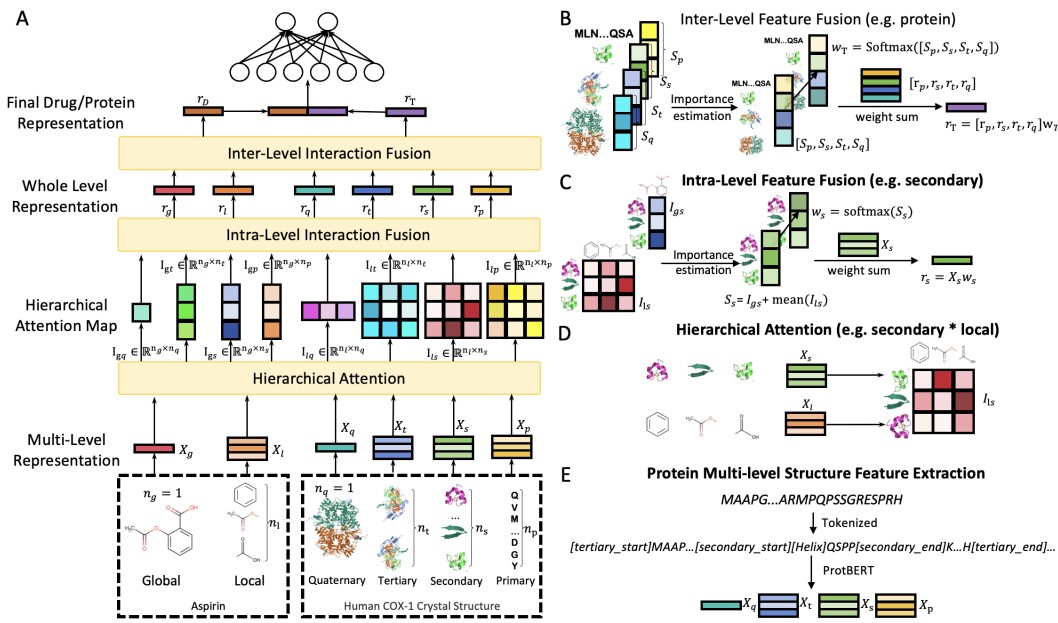

Figure 2: Overview of ColdDTI. (A) Overall architecture with hierarchical attention and fusion. (B) Inter-level feature fusion aggregates protein levels. (C) Intra-level feature fusion combines features within the same level. (D) Hierarchical attention captures drug–protein cross-level interactions. (E) Protein multi-level structure feature extraction with ProtBERT encoding.

## 4 METHODOLOGY

Inspired by the scientific insight about proteins' multi-level structures, we propose ColdDTI, a framework that captures interaction patterns between drug and multi-level structures of protein for cold-start DTI prediction. First, Section 4.1 describes how to extract structure feature of drug and protein, especially protein multi-level structure features according to available protein structure information. Then, Section 4.2 introduces hierarchical attention mechanism to mine hierarchical interaction patterns between drug and multi-level structures of protein. Finally Section 4.3 introduces feature fusion mechanism to leverage mined hierarchical interactions corresponding to different level structures for final DTI prediction.

### 4.1 PROTEIN MULTI-LEVEL STRUCTURE FEATURE EXTRACTION

As introduced in Section 3, protein $T = (a_1, a_2, ..., a_m)$ has multi-level structures. To extract feature of available protein multi-level structure information from biochemical database, we propose to expand the amino acid residue sequences (i.e. primary structure) usually used in prior works by inserting tags (e.g. [tertiary_start]) to starting and end point of secondary / tertiary structures, indicating their position (e.g. from the 100-th residue to the 200-th residue) and type (e.g $\alpha$-helix) on residue sequences. Then, we use pretrained protein transformer ProtTrans (Elnaggar et al., 2021) to extract protein multi-level structure features by adding these tags as special tokens to vocabulary of ProtTrans (Tai et al., 2020). With ProtTrans and its expanded vocabulary, we can obtain the dense representation of protein multi-level structures as $\mathbf{X}_p, \mathbf{X}_s, \mathbf{X}_t, \mathbf{X}_q$ for primary, secondary, tertiary and quaternary structure respectively. Specifically, the representation of each secondary or tertiary structure is calculated as the mean of all amino acid residue as well as special token representations in corresponding secondary or tertiary structure. More information of special tokens and implementation details are in Appendix A.

Similarly, we extract drug structure feature with pretrained drug transformer ChemBERTa-2 (Ahmad et al., 2022) to get $\mathbf{X}_l, \mathbf{X}_g$ for drug local and global structure respectively. Note that the representations at each structural level are learned through pretraining on abundant unlabeled drug and protein data, without relying on any interaction information.

## 4.2 HIERARCHICAL INTERACTIONS MINING

In this section, we introduce a hierarchical attention mechanism to capture the interactions between drug and protein. Existing methods primarily capture interactions between drug local structure and protein primary structures, limiting their ability to recognize influence from higher level protein structures to DTI result. Inspired by the success of hierarchical attention networks for modeling interactions across text of different granularities in natural language processing tasks (Yang et al., 2016; Lu et al., 2016), we then employ a hierarchical attention mechanism to model the interactions between each of two level drug structures (global and local) and each of four level protein structures (primary, secondary, tertiary, quaternary).

As an illustrative example, we model the interaction between a drug's local structure and a protein's secondary structure by using the drug representation $\mathbf{X}_l$ and the protein representation $\mathbf{X}s$ as inputs, and producing an interaction attention map $\mathbf{I}ls$ between the two structural levels. Specifically, $\mathbf{I}_{ls} = (\mathbf{W}_{ls}^l \mathbf{X}_l)(\mathbf{W}_{ls}^s \mathbf{X}_s)^\top$, where $\mathbf{W}_{ls}^l$ and $\mathbf{W}_{ls}^s$ are learnable parameters specific to interaction $\mathbf{I}_{ls}$ between the two level structures. The elements in the $i$-th row and $j$-th column of $\mathbf{I}_{ls}$ can represent the interaction intensity between the $i$-th local structure in drug molecule and the $j$-th secondary structure in protein.

Similarly, we calculate the interaction attention maps between drug and protein structures of other levels (e.g. between drug local structures and protein tertiary structures). More implementation details are in Appendix A.

## 4.3 HIERARCHICAL INTERACTION FOR REPRESENTATION FUSION

With hierarchical interaction attention maps obtained in Section 4.2, how to leverage these interactions across different level structures remains technical difficulty. Previous researches about molecular biology and proteomics suggest that chemical active structures are more inclined to interact with other structures, thereby usually having more important effect on DTI result (Schenone et al., 2013; Dudev & Lim, 2014). Inspired by this insight, we propose to estimate the importance of structures from each level to DTI according to mined hierarchical interaction attention map. Then, we use the estimated importance to fuse extracted multi-level structure representation into joint representation for final prediction.

Specifically, we design two-stage feature fusion process, where we first estimate importance of structures from the same level (e.g. all secondary structures in protein) to fuse their representations into a dense vector representing this whole level (e.g. fusing all representations in $\mathbf{X}_s$ into a dense vector representing secondary structure as a whole in protein), i.e. intra-level representation fusion. Then, we estimate importance of different levels of protein or drug so that we can fuse all level representations obtained during intra-level feature fusion into final representation of protein or drug, i.e. inter-level representation fusion.

**Intra-level representation fusion.** We take protein secondary structure as an example to illustrate intra-level representation fusion. According to Section 4.2, interaction attention maps related to protein secondary structures are $\mathbf{I}_{ls}$ with drug local structure and $\mathbf{I}_{gs}$ with drug global structure respectively, where the $j$-th column of the two interaction attention map represent the interaction intensity related to the $j$-th secondary structure in protein. We then calculate the interaction intensity of each secondary structure with $\mathbf{S}_s = (\mathbf{I}_{gs} + \mathbf{I}_{ls}.\text{m(axis=column)})$, where $.\text{m}(\cdot)$ calculate mean of matrix or vector. The $j$-th elements of $\mathbf{S}_s$ indicates the interaction intensity of $j$-th secondary structure with drug, including both drug global and local structures. Then, we apply $\text{Softmax}(\cdot)$ operator to $\mathbf{S}_s$ to normalize it into the importance weight of secondary structures as $\mathbf{w}_s = \text{Softmax}(\mathbf{S}_s)$. With this importance weight, we fuse the representations of all protein secondary structures into a dense vector to represent protein secondary structure as a whole as $\mathbf{r}_s = \mathbf{X}_s^\top \mathbf{w}_s$.

Similarly, we obtain each level structure representation as a whole, of protein (primary, tertiary and quaternary as $\mathbf{r}_p$, $\mathbf{r}_t$ and $\mathbf{r}_q$ respectively) and drug (local and global as $\mathbf{r}_l$ and $\mathbf{r}_g$ respectively). We show the implementation details in Appendix A.

**Inter-level representation fusion.** After getting the representation of each level structure as a whole, we fuse representations of different levels from the same side (i.e. drug side or protein side) to obtain representation of this side. Representations obtained in this way contain information of mined interaction and emphasize the feature of structures with stronger interaction intensity.

We take the protein multi-level structures as an example to illustrate inter-level feature fusion. Similar to the idea of intra-level representation fusion, the interaction intensity of each level protein structure indicates the importance of this level to DTI. Therefore, we calculate the mean of interaction intensities of all structures belonging to each level to get interaction intensity of this level as a whole. Then we apply $\mathrm{Softmax}(\cdot)$ to these intensities to get importance weight $\mathbf{w}_t$ of each level protein structure. Formally, $\mathbf{w}_T = \mathrm{Softmax}([\mathbf{S}_p.\mathrm{m}, \mathbf{S}_s.\mathrm{m}, \mathbf{S}_t.\mathrm{m}, \mathbf{S}_q.\mathrm{m}])$, where $\mathbf{S}_p, \mathbf{S}_t, \mathbf{S}_q$ are interaction intensities for primary, tertiary and quaternary structures respectively, the same as $\mathbf{S}_s$ for secondary structure described in intra-level feature fusion. With this importance weight, we can obtain the final representation of protein by $\mathbf{r}_T = [\mathbf{r}_p, \mathbf{r}_s, \mathbf{r}_t, \mathbf{r}_q]\mathbf{w}_T$.

Similarly, we obtain the final representation of drug as $\mathbf{r}_D$. The implementation details are in Appendix A. Finally, the two representations $\mathbf{r}_D$ and $\mathbf{r}_T$, are concatenated to form a joint representation, which is passed through a classification head implemented by multi-layer perception to generate the final prediction $\hat{y}$.

We use cross-entropy loss to train our model as $\mathcal{L}_{\mathrm{CE}} = -\mathbb{E}_{(D_i, T_i, y_i) \sim \mathcal{D}_{\mathrm{train}}}[y_i \log \hat{y} + (1 - y_i) \log(1 - \hat{y}_i)]$, where $\mathcal{D}_{\mathrm{train}}$ is training set containing known DTI samples. The trained model is then used to infer for novel drugs or proteins unseen in $\mathcal{D}_{\mathrm{train}}$.

## 5 EXPERIMENT

### 5.1 EXPERIMENTAL SETTINGS

**Dataset.** We evaluate our approach on four widely used benchmark datasets: **DrugBank** (Wishart et al., 2006), **BindingDB** (Liu et al., 2007), **BioSNAP** (Group, 2018), and **Human** (Tsubaki et al., 2019). The detailed statistics of these datasets are provided in Appendix B.1, Table 2. We consider three types of cold-start conditions: (i) *cold drug*, where training, validation, and test sets share no overlapping drugs while proteins are unrestricted; (ii) *cold protein*, where no proteins are shared across splits while drugs are unrestricted; and (iii) *cold pair*, where neither drugs nor proteins overlap across splits, ensuring no shared entities between training and test.

**Baselines.** We compare our proposed ColdDTI with a set of structure-based baselines, most of which are discussed in Section 2. The structure-based baselines can be grouped into two categories according to their feature encoders: **GNN-based drug feature encoders**, such as GraphDTA (Nguyen et al., 2020), which represent drugs as molecular graphs and use graph neural networks to learn topology-aware embeddings; and **Sequence-based feature encoders**, including MolTrans (Huang et al., 2021), TransformerCPI (Chen et al., 2020), HyperAttentionDTI (Zhao et al., 2022), Drug-BAN (Bai et al., 2023), and MlanDTI (Xie et al., 2024), which treat drugs and proteins as sequences and leverage attention or alignment mechanisms to capture interactions.

**Evaluation Metric.** DTI datasets are typically imbalanced, with far fewer positive interactions than negative ones. In such cases, overall accuracy can be misleading, as a model that simply predicts the majority class would still achieve a high score. Therefore, we focus on metrics that better reflect the ability to correctly identify both positive and negative classes. Specifically, we adopt the AUC, the AUPR and the F1 score. AUC measures the overall discriminating ability across thresholds, AUPR is particularly suitable for imbalanced data as it emphasizes the quality of positive predictions, and F1, as the harmonic mean of precision and recall, balances the trade-off between capturing true positives and avoiding false positives. Together, these metrics provide a comprehensive evaluation of model performance under the class-imbalance characteristics of DTI prediction.

### 5.2 PERFORMANCE COMPARISON

From Table 1, we can observe that our proposed ColdDTI achieves almost all the best performance on these metrics across all three cold-start settings (cold pair, cold drug, and cold protein).

In **cold drug** setting, most methods can achieve relatively good performance compared with other 2 settings, which makes it difficult to distinguish one dominate method. However, ColdDTI still achieves the best or second-best. On BindingDB, some existing methods already perform well, such as DrugBAN (0.886), while ColdDTI further improves the AUC to 0.896. On BioSNAP, all baselines achieving AUCs between 0.81–0.83, and ColdDTI reaching the second-best result (0.832). On

Table 1: Test performance on BindingDB, BioSNAP, Human, and DrugBank datasets under cold drug, cold protein, and cold pair settings. The best performance is highlighted in **bold**, while the second-best performance is marked with underline. All of the results are mean of 3 random runs.

| Dataset | Methods | Cold Drug | | | Cold Protein | | | Cold Pair | | |
|---|---|---|---|---|---|---|---|---|---|---|
| | | AUC | AUPR | F1 | AUC | AUPR | F1 | AUC | AUPR | F1 |
| BindingDB | GraphDTA | .819 | .834 | .729 | .628 | .453 | .281 | .582 | .531 | .508 |
| | MolTrans | .839 | .826 | .741 | .617 | .445 | .502 | .595 | .522 | .511 |
| | TransformerCPI | .826 | .838 | .738 | .695 | .567 | .550 | .656 | .594 | .566 |
| | HyperAttDTI | .875 | .847 | .759 | .671 | .511 | .514 | .661 | .598 | .582 |
| | DrugBAN | .886 | **.865** | .768 | .609 | .462 | .352 | .655 | .600 | .542 |
| | MlanDTI | .848 | .851 | .747 | .739 | .540 | .596 | .671 | .594 | .601 |
| | ColdDTI | **.896** | .861 | **.775** | **.751** | **.579** | **.609** | **.742** | **.652** | **.634** |
| BioSNAP | GraphDTA | .815 | .812 | .706 | .723 | .746 | .641 | .703 | .694 | .557 |
| | MolTrans | .824 | .823 | .726 | .744 | .771 | .664 | .681 | .693 | .528 |
| | TransformerCPI | .825 | .828 | .713 | .765 | .757 | .645 | .728 | .746 | .634 |
| | HyperAttDTI | .811 | .811 | .712 | .789 | .817 | .669 | .778 | .783 | .587 |
| | DrugBAN | **.835** | .830 | .729 | .678 | .699 | .443 | .660 | .636 | .562 |
| | MlanDTI | .824 | .827 | .719 | .841 | **.868** | .735 | .782 | **.801** | .653 |
| | ColdDTI | .832 | **.833** | **.733** | **.847** | .867 | **.759** | **.791** | .798 | **.695** |
| Human | GraphDTA | .934 | .953 | .872 | .783 | .787 | .702 | .446 | .599 | .140 |
| | MolTrans | .913 | .934 | .850 | .713 | .534 | .561 | .720 | .815 | .516 |
| | TransformerCPI | .931 | .963 | .874 | .832 | .809 | .734 | .675 | .775 | .471 |
| | HyperAttDTI | .940 | .958 | .821 | .859 | .812 | .798 | .788 | .836 | .536 |
| | DrugBAN | **.951** | .962 | .886 | .817 | .805 | .724 | .782 | .842 | .616 |
| | MlanDTI | .944 | .961 | .866 | .859 | .817 | .801 | .794 | .833 | .571 |
| | ColdDTI | .947 | **.964** | **.889** | **.864** | **.824** | **.810** | **.818** | **.847** | **.676** |
| DrugBank | GraphDTA | .816 | .807 | .718 | .517 | .569 | .497 | .497 | .492 | .456 |
| | MolTrans | .809 | .799 | .675 | .692 | .755 | .576 | .531 | .535 | .246 |
| | TransformerCPI | .760 | .774 | .658 | .650 | .691 | .293 | .547 | .553 | .323 |
| | HyperAttDTI | .818 | .814 | .710 | .759 | .812 | .631 | .504 | .512 | .187 |
| | DrugBAN | .829 | .823 | .730 | .718 | .756 | .526 | .515 | .502 | .144 |
| | MlanDTI | .836 | .814 | .706 | .807 | .844 | .730 | .540 | .532 | .205 |
| | ColdDTI | **.849** | **.851** | **.760** | **.837** | **.864** | **.749** | **.583** | **.591** | **.584** |

Human, most methods achieve high performance with AUCs above 0.93, indicating that this dataset is inherently easier to model; ColdDTI ranks second-best in this case. On DrugBank, ColdDTI attains the best AUC of 0.849, while also outperforming all baselines in terms of AUPR and F1.

The **cold protein** setting is generally more challenging than cold drug. The representation space of proteins themselves is higher-dimensional and more complex. In cold drug scenario, the model can rely on shared proteins to learn stable binding patterns. However, in cold protein scenario, the complex and brand-new proteins make the model lack transfer references, thus posing greater challenges. Although the setting is more difficult, ColdDTI achieves the best among all the baselines and datasets. On BindingDB, most methods fail to reach an AUC of 0.7, whereas ColdDTI achieves an AUC of 0.751 and an F1 of 0.609, significantly outperforming the baselines. On BioSNAP, Cold-DTI attains the best AUC 0.847. On Human, ColdDTI achieves an AUC of 0.864, outperforming MlanDTI and HyperAttentionDTI (both 0.859), showing stronger generalization. On DrugBank, ColdDTI consistently delivers the best results across all metrics, with a notable margin over the baselines.

The **cold pair** setting is the most challenging, as neither drugs nor proteins overlap between training and testing, requiring the model to capture truly transferable interaction patterns. On BindingDB, all methods fail to reach an AUC of 0.7, while ColdDTI outstands significantly with an AUC of 0.742 and an F1 of 0.634. On BioSNAP, ColdDTI again outperforms all baselines, achieving an AUC of 0.791 and an F1 of 0.695. On Human, all methods experience a sharp performance drop in the cold pair setting, for instance, GraphDTA's AUC falls to only 0.446, while ColdDTI still achieves an AUC of 0.818. On DrugBank, most methods are close to random (AUC around 0.50–0.55), highlighting

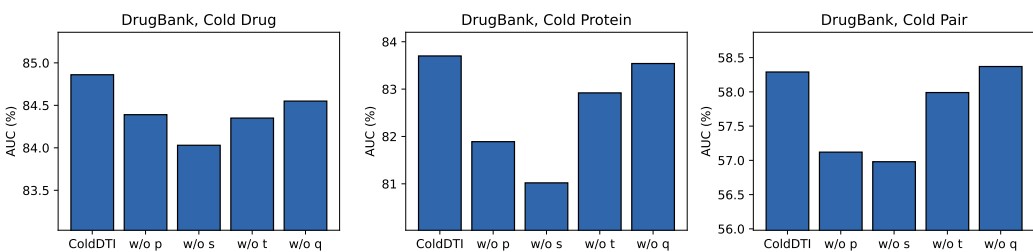

Figure 3: Ablation study on the DrugBank dataset under cold drug, cold protein, and cold pair settings.

the extreme difficulty of this dataset in the cold pair scenario. Nevertheless, ColdDTI still maintains state-of-the-art performance, boosting the AUC from below 0.55 to 0.583, and achieving an F1 of 0.584, demonstrating robust cross-distribution generalization.

Overall, the cold pair setting is the most difficult, yet ColdDTI consistently achieves substantial improvements across all datasets. These results confirm that ColdDTI, by modeling multi-level protein structures and cross-level interactions, effectively captures biologically transferable patterns and achieves comprehensive and stable performance gains in cold-start DTI prediction.

## 5.3 ABLATION STUDY

To demonstrate the effectiveness of hierarchical interactions mined in Section 4.2, we design four variants of ColdDTI by removing interactions corresponding to four level protein structures respectively, and compare their performance with ColdDTI. Specifically, we design the following variants:

- **w/o p**: without interaction attention map corresponding to primary structure, i.e. $\mathbf{I}_{lp}$ and $\mathbf{I}_{gp}$.
- **w/o s**: without interaction attention map corresponding to secondary structure, i.e. $\mathbf{I}_{ls}$ and $\mathbf{I}_{gs}$.
- **w/o t**: without interaction attention map corresponding to tertiary structure, i.e. $\mathbf{I}_{lt}$ and $\mathbf{I}_{gt}$.
- **w/o q**: without interaction attention map corresponding to quaternary structure, i.e. $\mathbf{I}_{lq}$ and $\mathbf{I}_{gq}$.

The implementation details of the four variants are in Appendix B.3. We evaluate the four variants on DrugBank dataset across three cold-start settings. Figure 3 shows the results.

As shown in Figure 3, we can observe that ColdDTI surpasses all other four variants, demonstrating the effectiveness of hierarchical interactions mining in Section 4.2. Compared with ColdDTI, **w/o p**, **w/o s** and **w/o t** perform apparently worse than to ColdDTI, highlighting the effectiveness of interactions related to these three level protein structures in our framework on the whole. However, ColdDTI has almost no advantage over **w/o q**, suggesting that interactions about quaternary level may contribute little to the overall performance.

Specifically, the performance of **w/o p** decreases obviously compared with ColdDTI, demonstrating that the primary structure plays important role in DTI result and supporting previous works (Huang et al., 2021; Zhao et al., 2022; Bai et al., 2023) that represent protein as amino acid residue sequence (i.e. primary structure). However, **w/o s** and **w/o t** also perform worse than ColdDTI, showing that whether the potential interaction can be triggered is influenced by higher level (secondary and tertiary) structure, due to factors like spatial structure posed by these higher level structures, verifying the effectiveness of implementing protein multi-level structure-aware interaction. **w/o t** performs better than **w/o s**, showing that secondary structure may play more important role for used Drugbank dataset than tertiary structure.

The most noticeable performance drop occurs in the cold protein setting, indicating that mining interactions corresponding to protein multi-level structure is especially effective when predicting DTI for unseen proteins.

The performance of **w/o q** is very close to that of ColdDTI, differing from other three variants. This suggests that the interaction mined about protein quaternary structure contributes little to the overall performance. This result is reasonable since proteins are very large with long amino acid sequences, while drug molecules are typically small. Additionally, biologically active proteins usually have

specific binding sites for drug interactions. These properties make it difficult to represent proteins as a whole to interact with drugs at the quaternary structure level.

### 5.4 CASE STUDY

We visualize two interaction attention map ($\mathbf{I}_{lp}$ and $\mathbf{I}_{ls}$) of a well-researched DTI case (DB00945 and P23219) in Figure 4, output by ColdDTI trained on DrugBank dataset, to show ColdDTI can capture explainable influence of structures from different levels to DTI results.

Figure 4 shows the process of -COO- group of DB00945 (drug) interacting with -OH group of 530-th Ser (referred to as Ser-530, the same below) in P23219 (protein) the 140-th $\alpha-$helix (referred to as helix-140, the same below) (Rouzer & Marnett, 2020). In both maps, -COOH and -COO- group in drug shows stronger interaction intensity, complying with their relatively more active property than other two groups. According to the map involving protein primary structure, ColdDTI give high attention to interaction between -COO- and Ser-530, accurately capturing the functional group and amino acid residue involved in this interaction. ColdDTI also captures the secondary structures that contribute to this results. In map about protein secondary structure, ColdDTI gives high attention to interaction between -COO- and helix-140, where Ser-530 located. Moreover, ColdDTI also captures other secondary structures that contribute to DTI result (a structure on the left of helix-140 in red box and one between 40 an 60 have relatively high interaction intensity). These secondary structures do not interact with DB00945 directly but provide stable skeleton and help to adjust spatial direction of Ser530 to support interaction (Lucido et al., 2016; Lei et al., 2015).

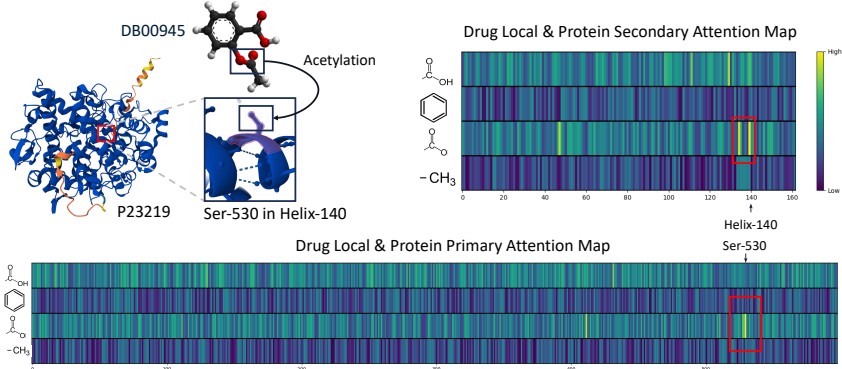

Figure 4: Case study of -COO- group of DB00945 interacts with -OH group of Ser-530 in P23219.

## 6 CONCLUSION

We introduce ColdDTI, a novel framework designed to tackle challenges of cold-start drug-target interaction (DTI) prediction. By leveraging hierarchical interaction patterns and dynamically estimating substructure importance weights based on these patterns, our method captures the complex dynamics of drug-protein interactions more effectively. Extensive experiments across multiple benchmark datasets and cold-start settings demonstrate that ColdDTI outperforms existing methods, particularly in cold-start protein settings.

**Limitations.** While ColdDTI presents a promising approach, there are some limitations that need to be addressed in future work. (1) Current approach relies on structural information to simulate cold-start settings, but in practice, multimodal data, such as morphological or chemical property information, may offer additional insights and benefits. Future efforts will focus on exploring the integration of multimodal data sources to further enhance cold-start DTI prediction performance. These improvements will bring us closer to achieving more accurate, generalizable, and efficient solutions for AI-driven drug discovery.

## REPRODUCIBILITY STATEMENT

We give the implementation details of ColdDTI in Appendix A. We open the source code of ColdDTI with an anonymous repository as `https://anonymous.4open.science/r/Code-ColdDTI-AC54`.

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

# A  IMPLEMENTATION DETAILS OF COLDDTI

## A.1  IMPLEMENTATION DETAILS FOR MULTI-LEVEL STRUCTURE FEATURE EXTRACTION

We have the following tags as special tokens to indicate positions or types of multi-level structure.

- `[secondary_start]`, indicating the start position of a secondary structure.
- `[secondary_end]`, indicating the start position of a secondary structure.
- `Helix`, `Sheet`, `Turn` and `Bend` are put after `[secondary_start]`, indicating the type of secondary structures.
- `[tertiary_start]`, ndicating the start position of a secondary structure.
- `[tertiary_end]`, ndicating the start position of a secondary structure.

We do not set tags to indicate quaternary structure because quaternary structure is usually the global structure of proteins. For representations of each level, we calculate the mean of representations belong to this level. For example, a secondary structure $\alpha$-helix starts from the 100-th residue and ends at 200-th residue, the we calculate the mean of representations for , `[secondary_start]`, `Helix`, the residues in this structures and `[secondary_end]` as the representation for this $\alpha$-helix.

## A.2  IMPLEMENTATION DETAILS FOR HIERARCHICAL INTERACTIONS

In Section 4.2, we have introduced how to calculate the interaction attention map $\mathbf{I}_{ls}$ between drug local structure and protein secondary structure. The interaction attention map involving other levels is calculated as follows.

**Drug local & protein primary.** $\mathbf{I}_{lp} = (\mathbf{W}_{lp}^l \mathbf{X}_l)(\mathbf{W}_{lp}^p \mathbf{X}_p)^\top$, where $\mathbf{W}_{lp}^l$ and $\mathbf{W}_{lp}^p$ are learnable parameters.

**Drug local & protein tertiary.** $\mathbf{I}_{lt} = (\mathbf{W}_{lt}^l \mathbf{X}_l)(\mathbf{W}_{lt}^t \mathbf{X}_t)^\top$, where $\mathbf{W}_{lt}^l$ and $\mathbf{W}_{lt}^t$ are learnable parameters.

**Drug local & protein quaternary.** $\mathbf{I}_{lq} = (\mathbf{W}_{lq}^l \mathbf{X}_l)(\mathbf{W}_{lq}^q \mathbf{X}_q)^\top$, where $\mathbf{W}_{lq}^l$ and $\mathbf{W}_{lq}^p$ are learnable parameters.

**Drug global & protein primary.** $\mathbf{I}_{gp} = (\mathbf{W}_{gp}^g \mathbf{X}_g)(\mathbf{W}_{gp}^p \mathbf{X}_p)^\top$, where $\mathbf{W}_{gp}^l$ and $\mathbf{W}_{gp}^p$ are learnable parameters.

**Drug global & protein secondary.** $\mathbf{I}_{gs} = (\mathbf{W}_{gs}^g \mathbf{X}_g)(\mathbf{W}_{gs}^s \mathbf{X}_s)^\top$, where $\mathbf{W}_{gs}^g$ and $\mathbf{W}_{gs}^s$ are learnable parameters.

**Drug global & protein tertiary.** $\mathbf{I}_{gt} = (\mathbf{W}_{gt}^l \mathbf{X}_g)(\mathbf{W}_{gt}^t \mathbf{X}_t)^\top$, where $\mathbf{W}_{gt}^g$ and $\mathbf{W}_{gt}^t$ are learnable parameters.

**Drug global & protein quaternary.** $\mathbf{I}_{gq} = (\mathbf{W}_{gq}^g \mathbf{X}_g)(\mathbf{W}_{gq}^q \mathbf{X}_q)^\top$, where $\mathbf{W}_{gq}^g$ and $\mathbf{W}_{gq}^q$ are learnable parameters.

## A.3  IMPLEMENTATION DETAILS FOR REPRESENTATION FUSION

In the main text (Section 4.3), we described the two-stage feature fusion framework. Here, we provide additional implementation details.

### A.3.1  INTRA-LEVEL FEATURE FUSION

For each layer of the protein structure (primary, secondary, tertiary, quaternary), they fuse in the same way.

- **Primary.** $\mathbf{S}_p = \mathbf{I}_{gp} + \mathbf{I}_{lp}.\mathrm{m}(\mathrm{axis=column})$, $\mathbf{w}_p = \mathrm{Softmax}(\mathbf{S}_p)$, $\mathbf{r}_p = \mathbf{X}_p^\top \mathbf{w}_p$.
- **Secondary.** $\mathbf{S}_s = \mathbf{I}_{gs} + \mathbf{I}_{ls}.\mathrm{m}(\mathrm{axis=column})$, $\mathbf{w}_s = \mathrm{Softmax}(\mathbf{S}_s)$, $\mathbf{r}_s = \mathbf{X}_s^\top \mathbf{w}_s$.
- **Tertiary.** $\mathbf{S}_t = \mathbf{I}_{gt} + \mathbf{I}_{lt}.\mathrm{m}(\mathrm{axis=column})$, $\mathbf{w}_t = \mathrm{Softmax}(\mathbf{S}_t)$, $\mathbf{r}_t = \mathbf{X}_t^\top \mathbf{w}_t$.

- **Quaternary.** $\mathbf{S}_q = \mathbf{I}_{gq} + \mathbf{I}_{lq}.\mathrm{m}(\mathrm{axis=column})$, $\mathbf{r}_q = \mathbf{X}_q^\top \mathbf{S}_q$.

The local/global processing of drugs is handled in the same way:

- **Local.** $\mathbf{S}_l = \mathbf{I}_{lp}.\mathrm{m}(\mathrm{axis=row}) + \mathbf{I}_{ls}.\mathrm{m}(\mathrm{axis=row}) + \mathbf{I}_{lt}.\mathrm{m}(\mathrm{axis=row}) + \mathbf{I}_{lq}.\mathrm{m}(\mathrm{axis=row})$, $\mathbf{w}_l = \mathrm{Softmax}(\mathbf{S}_l)$, $\mathbf{r}_l = \mathbf{X}_l^\top \mathbf{w}_l$.
- **Global.** $\mathbf{S}_g = \mathbf{I}_{gp}.\mathrm{m}(\mathrm{axis=row}) + \mathbf{I}_{gs}.\mathrm{m}(\mathrm{axis=row}) + \mathbf{I}_{gt}.\mathrm{m}(\mathrm{axis=row}) + \mathbf{I}_{gq}.\mathrm{m}(\mathrm{axis=row})$, $\mathbf{r}_g = \mathbf{X}_g^\top \mathbf{w}_g$.

### A.3.2 INTER-LEVEL FEATURE FUSION

After obtaining the representation of each level as a whole, we further fuse different levels from the same side (drug or protein) into a final side-specific representation.

- **Protein.** For protein, the interaction intensity of each level is computed as the mean of its intra-level scores.
$$\mathbf{w}_T = \mathrm{Softmax}([\mathbf{S}_p.\mathrm{m}, \mathbf{S}_s.\mathrm{m}, \mathbf{S}_t.\mathrm{m}, \mathbf{S}_q.\mathrm{m}]),$$
where $\mathbf{S}_p, \mathbf{S}_s, \mathbf{S}_t, \mathbf{S}_q$ are the interaction intensity vectors of primary, secondary, tertiary and quaternary structures. With these weights, the final protein representation is
$$\mathbf{r}_T = [\mathbf{r}_p, \mathbf{r}_s, \mathbf{r}_t, \mathbf{r}_q]\,\mathbf{w}_T.$$

- **Drug.** Similarly, for drugs we combine local and global representations by:
$$\mathbf{w}_D = \mathrm{Softmax}([\mathbf{S}_l.\mathrm{m}, \mathbf{S}_g.\mathrm{m}]),$$
where $\mathbf{S}_l$ and $\mathbf{S}_g$ are the interaction intensity vectors of local and global drug structures. The final drug representation is
$$\mathbf{r}_D = [\mathbf{r}_l, \mathbf{r}_g]\,\mathbf{w}_D.$$

Finally, the joint representation is formed by concatenating $\mathbf{r}_D$ and $\mathbf{r}_T$, i.e.
$$\mathbf{z} = [\mathbf{r}_D; \mathbf{r}_T],$$
which is passed through a multi-layer perceptron classifier to generate the final prediction $\hat{y}$.

## B EXPERIMENT DETAILS

### B.1 IMPLEMENTATION DETAILS

We implement ColdDTI in PyTorch and conduct all experiments on a single NVIDIA RTX 3090 GPU with 24GB memory. The model is trained using the Adam optimizer with an initial learning rate of $5 \times 10^{-5}$, weight decay of $1 \times 10^{-4}$, and a batch size of 64. We train for up to 20 epochs with early stopping based on validation performance. The learning rate is decayed by a factor of 0.5 every 5 epochs. All the other hyperparameters follow the default settings unless otherwise specified.

### B.2 DATASET DETAILS

The number of drugs, proteins and drug-target interactions (positive and negative) of four datasets are summarized in Table 2.

| Datasets | Drug | Protein | Positive | Negative |
|---|---|---|---|---|
| BindingDB | 14643 | 2623 | 20764 | 28525 |
| Human | 2726 | 2001 | 3364 | 3364 |
| BioSNAP | 4505 | 2181 | 13830 | 13634 |
| Drugbank | 6643 | 4252 | 17511 | 17511 |

Table 2: Summary of benchmark datasets.

We classify the cold start scenarios into the following three specific splits:

- *Cold drug.* $\mathcal{D}_{\text{train}}$, $\mathcal{D}_{\text{val}}$ and $\mathcal{D}_{\text{test}}$ contain no overlapping drugs, with no restriction on the target proteins. The dataset is split by drugs in an 8:1:1 ratio, ensuring that the drugs across these three subsets are mutually exclusive.

- *Cold protein.* $\mathcal{D}_{\text{train}}$, $\mathcal{D}_{\text{val}}$ and $\mathcal{D}_{\text{test}}$ contain no overlapping proteins, with no restriction on the drug sets. Similarly, the dataset is split by proteins in an 8:1:1 ratio.

- *Cold pair.* Both drug and protein sets in the training, validation, and test splits have no overlap, satisfying the no-intersection requirement for both drugs and proteins. Concretely, we first split the drugs in an 8:1:1 ratio with no overlap, then split the proteins in the same ratio. If no interaction exists for a drug–protein combination in the resulting subsets, the pair is discarded.

### B.3    BASELINE DETAILS

All the baseline methods, GraphDTA[1] (Nguyen et al., 2020), Moltrans[2] (Huang et al., 2021), TransformerCPI[3] (Chen et al., 2020), HyperAttDTI[4] (Zhao et al., 2022), DrugBAN[5] (Bai et al., 2023) and MlanDTI[6] (Xie et al., 2024) are implemented according to the open-source code attached with the original paper and employed with their default configurations as specified by the authors. Specifically, since we don't conduct cross-domain evaluation in our experiment, we select vanilla DrugBAN without CDAN module and MlanDTI without pseudo labeling. For GraphDTA, it has four variants of GNN to extract drug features. We choose GAT_GCN to extract drug feature according to the reports provided in the original paper.

### B.4    CASE STUDY DETAILS

ColdDTI used in case study is trained with the DrugBank dataset with hyperparameters tuned on DrugBank validation dataset under cold pair setting. The three cases are selected from BioSNAP database, ensuring that the selected drugs and proteins are all unseen in DrugBank dataset. The drug local structure granularity processed by ColdDTI is smaller than the local structures showed in Figure 4, thus there are too many local structures, making it not convenient to illustrate in Figure 4. Therefore, we sum the importance of processed drug local structures to form the importance of structures in Figure 4 for convenience.

## C    USE OF LARGE LANGUAGE MODELS (LLMS)

In this work, we used a large language model merely as an assistant for polishing the writing of the manuscript (e.g., grammar refinement). All technical content, experimental design, and analysis were conceived, implemented, and validated entirely by the authors. All outputs from the LLM were manually checked, edited, and verified by the authors before inclusion in the paper.

---

[1]https://github.com/thinng/GraphDTA
[2]https://github.com/kexinhuang12345/moltrans
[3]https://github.com/lifanchen-simm/transformerCPI
[4]https://github.com/zhaoqichang/HpyerAttentionDTI
[5](https://github.com/peizhenbai/DrugBAN
[6]https://github.com/CMACH508/MlanDTI

