# OpenReview forum: "Attending on Multilevel Structure of Proteins enables Accurate Prediction of Cold-Start Drug-Target Interactions"
_ICLR.cc/2026/Conference — Submitted to ICLR 2026_

### Official Review · Reviewer_FX9M · 2025-10-26

**Soundness:** 2
**Presentation:** 2
**Contribution:** 1
**Rating:** 2
**Confidence:** 5

**Summary:**

The paper presents ColdDTI, a framework that models proteins’ multi-level structures to improve cold-start drug–target interaction prediction. Using hierarchical attention and adaptive fusion, it captures cross-level drug–protein interactions. Experiments on multiple benchmarks show consistent gains over baselines.

**Strengths:**

The paper presents a strong motivation, as cold-start DTI prediction remains an urgent and important challenge in the field.

**Weaknesses:**

1. The proposed multi-level structure framework lacks sufficient novelty. Similar hierarchical or multi-level attention mechanisms have been explored in prior works, such as MHADTI (a) and MlanDTI (b)

Reference:

(a) MHADTI: Predicting Drug–Target Interactions via Multiview Heterogeneous Information Network Embedding with Hierarchical Attention Mechanisms (Briefings in Bioinformatics, 2022)

(b) Multilevel Attention Network with Semi-supervised Domain Adaptation for Drug–Target Prediction (AAAI, 2024).

2. The experimental comparison is incomplete. The paper does not include several recent strong baselines that specifically address cold-start DTI prediction, such as GraphBAN (a) and FusionDTI (b).

​​Reference:

(a) GraphBAN: An inductive graph-based approach for enhanced prediction of compound-protein interactions (Nature Communications, 2025).

(b) FusionDTI: Fine-grained Binding Discovery with Token-level Fusion for Drug-Target Interaction (EMNLP 2025).

3.  The paper does not provide sufficient analysis or explanation for the performance drop observed under cold-start settings.
The experiments are conducted on several cold-start datasets, but the paper does not clarify why performance drops in certain settings whether the degradation arises from technical limitations of the framework, intrinsic data sparsity, or the nature of the cold-start split itself.

4. The method claims to capture biologically meaningful multi-level protein structures, but the explainability analysis (attention visualisations) remains superficial.
It would be beneficial to quantitatively validate whether the learned attention indeed aligns with known structural or functional regions in proteins.

5.  Some sections are described in a verbose or repetitive descriptions, making it difficult to follow the methodological flow (e.g., hierarchical interaction).

**Questions:**

Please refer to the above weaknesses for detailed comments and suggestions.

---

### Official Review · Reviewer_znRg · 2025-10-29

**Soundness:** 3
**Presentation:** 2
**Contribution:** 3
**Rating:** 6
**Confidence:** 3

**Summary:**

This paper presents a new approach to predict drug-target interactions (DTI) in the cold-start scenario (i.e., for newly developed drugs or/and new proteins) based on
 1. presenting drugs and proteins at different structural levels (local vs global levels for drugs and primary, secondary and tertiary levels for proteins),
 2. learning and capturing interactions at between the drug sub-structures and protein sub-structures, and
 3. fusing the representation of the different sub-structures in a single vector representation based on their interaction strength.

**Strengths:**

1) The paper tackles an important problem in the drug discovery process,  predicting drug-target interactions. Contrary to other works, it focuses on the more realistic and challenging cases of making predictions about new drugs or new proteins.
2) To the best of my knowledge, the proposed multilevel structure representation is very novel and interesting.
3) The comprehensive experimental evaluation over 4 well-known DTI datasets show that the proposed approach clearly outperforms many state-of-the-art approaches on three key metrics (AUC, AUPR and F1-score)

**Weaknesses:**

Although I believe that the proposed approach is technically sound, the technical content is poorly presented in section 4.

   a. A lot of matrix multiplications do not make sense because they violate the basic matrix multiplication size rule (i.e., the number of columns in the first matrix must equal the number of rows in the second matrix). For example, although not explicitly stated, one can infer that the matrix X_l of local structure embeddings of the drug is a d x l matrix (where d is the embedding size and l is the number of local structures). Likewise, X_s is a d x s matrix, where d is the embedding size and s is the number of secondary structures of the protein. With those matrix dimensions,   I_{ls} = (W^l_{ls}Xl_)(W^s_{ls}X_s)^⊤ is incorrect. It should be replaced with  I_{ls} = (W^l_{ls}Xl_)^⊤ (W^s_{ls}X_s) .  A similar issue is present in the definition of rs = (X_s)^⊤ w_s. It should be   rs = X_s (w_s)^⊤

   b. The formula defining the intra-level representation fusion does not make sense: S_s = (I_{gs} + I_{ls}.m(axis=column)). It adds two matrices with incompatible sizes:
      i) the g x s matrix  I_{gs}  of interactions between g global drug structures and s protein secondary structures, and
     ii) the 1 x s matrix  I_{ls}.m(axis=column) of the scalar strengths of the s secondary protein structures
  It should be replaced with S_s = I_{gs}.m(axis=column) + I_{ls}.m(axis=column)

**Questions:**

N/A

---

### Official Review · Reviewer_dbpi · 2025-10-31

**Soundness:** 2
**Presentation:** 3
**Contribution:** 2
**Rating:** 4
**Confidence:** 4

**Summary:**

This paper proposes a hierarchical attention network that leverages multi-level protein structural representations to address the limited prediction performance of existing DTI models on unseen drugs, targets, and drug–target pairs. The unseen prediction task is analogous to the cold-start problem in recommendation systems, which is an interesting and novel perspective in the context of DTI prediction. The authors employ pretrained models ProtTrans and ChemBERTa-2 to extract multi-level representations of proteins and compounds, same as a previous approache named MlanDTI. By effectively capturing and integrating interaction and fusion information across these hierarchical representations, the proposed model, ColdDTI, achieves superior performance compared to current state-of-the-art methods. Furthermore, the ablation experiments convincingly demonstrate the importance of each level feature representation in contributing to the overall predictive performance.

**Strengths:**

1.	Effectively targets the critical cold-start DTI challenge, where generalization to novel drugs/proteins is essential, by learning transferable structural interaction patterns rather than relying on interaction-specific data.

2.	Reasonably extends protein representation to include higher-level structures (secondary to quaternary) using special tokens in ProtBERT, offering a more comprehensive view than primary-sequence-only approaches.

3. The overall language and methodological presentation of the paper is clear and well-organized.

**Weaknesses:**

1. Potential overfitting to benchmarks: maybe the paper doesn't address dataset-specific biases (e.g., over-representation of certain protein families). Under cold pair setting, even if the proteins in test set don’t appear in training set, if the proteins in test set belong to the same family with those in training set, the results may not sufficiently reflect the generalization ability. So if possible, it is recommended to supplement an experiment under the setting of cold protein family, namely the protein families in training set, validation set and test set without intersection.

2. Insufficient baselines: while structure-based baselines are covered, graph-based methods are dismissed early without direct comparison in cold-start settings. Recent works like EviDTI (using 2D/3D drug info) are mentioned but not benchmarked against. It is suggested to add one or two graph-based methods along with EviDTI to baselines, which can make the results more convinced.

3. Lack of ablation study for drug representation: only structural levels of proteins are discussed in ablation study, so the ablation study can be extended to drug representations by examining the individual contributions of the global and local structures, which would provide a more comprehensive validation of the proposed framework.

4. Insufficient discussion related to prediction results: sometimes the results in four datasets variate a great deal, especially under cold pair setting (Table 1). So what cause the difference among them? It is suggested to analyze the difference of protein families and molecular types in four datasets, which can provide valuable biological insights.

5. Although unseen prediction is important, focusing solely on this aspect makes the study somewhat limited, as it lacks evaluation on seen drugs and proteins, which are also essential for a complete DTI benchmark.

**Questions:**

1. With recent models like SaProt and ESM-IF1 directly incorporating 3D structural contexts (e.g., inverse folding or residue-pair distances) into protein embeddings, what specific benefits does your sequence-based tag insertion (e.g., [tertiary_start]) offer in terms of simplicity, alignment with SMILES-drug modeling, or performance in cold-protein generalization? In addition,  if structure are already available, more accurate structure-based methods—such as molecular docking—might be more suitable and informative.

2. The ablation study seems to indicate that quaternary structure is not very important in DTI prediction, but is it because that spatial structural information is not integrated in quaternary representation? If integrate 3D features from tools like AlphaFold for proteins with available structures, would the results be further improved?

3.  Leveraging pretrained model knowledge to address the cold-start problem is a common and effective strategy, since the pretrained representations may already capture implicit information related to unseen entries. In Table 1, only MlanDTI employs large-scale pretraining, which may explain its overall second-best performance. Therefore, it would be more convincing to include additional pretrained-model-based DTI methods for comparison to better demonstrate the superiority of ColdDTI. Furthermore, if possible, the authors are encouraged to analyze the contribution of using pretrained representations compared to direct feature embeddings.

4. The case study is too limited in scope. To enhance model interpretability, it is recommended to include feature-space visualizations (e.g., t-SNE or PCA plots) that separate positive and negative samples, thereby providing deeper insights into how the model distinguishes binding and non-binding pairs.

---

### Official Review · Reviewer_p6sz · 2025-11-01

**Soundness:** 2
**Presentation:** 2
**Contribution:** 2
**Rating:** 4
**Confidence:** 2

**Summary:**

The ColdDTI paper presents an attention-based model for predicting how drugs interact with proteins, even when the model has never seen those drugs or proteins before. Unlike older models that only use protein sequences or molecular graphs, ColdDTI uses multiple protein structure levels from amino acid sequences to 3D folding and connects them with local and overall features of drugs. It builds cross-attention maps between these levels, such as drug fragments and protein 3D structures, and combines them through a two-step fusion to decide if a drug and protein will interact. The model performs well across four datasets, especially for unseen proteins or drug–protein pairs. However, the authors used their own data split instead of stricter testing methods like cluster-start or bias-reduced splits, and they did not compare results with recent top-performing models such as EviDTI (2025) or ColdstartCPI (2025), so it is hard to tell how much better their method really is.

**Strengths:**

* The model explicitly incorporates multi-level protein structure and local/global drug representations for richer biochemical context.
* It demonstrates verified performance gains through multi-level cross-attention and two-stage fusion.
* Attention maps allow visualization of residue–fragment contributions to interaction predictions.

**Weaknesses:**

* The paper does not include direct benchmarks against strong contemporary SOTAs like EviDTI [1] and ColdstartCPI [2], leaving uncertainty about true performance standing.
* The paper uses a self-defined 8:1:1 split instead of existing splits employing cold-start or cluster-start splits with ligand-bias reduction preprocessing [3], which weakens the statistical rigor of its “true cold-start” claims.


 - [1] Zhao, Yanpeng, et al. "Evidential deep learning-based drug-target interaction prediction." Nature Communications 16.1 (2025): 6915.
 - [2] Zhao, Qichang, et al. "ColdstartCPI: Induced-fit theory-guided DTI predictive model with improved generalization performance." Nature Communications 16.1 (2025): 6436.
 - [3] https://github.com/Lzcstan/DrugLAMP

**Questions:**

Have you evaluated or planned to compare ColdDTI with more recent state-of-the-art models such as EviDTI (2025) or ColdstartCPI (2025) to better position your model’s performance in the current landscape?

---

### Meta-Review · Area_Chair_bm1v · 2026-01-08

**Summary:**

Reviewers generally found the paper to be targeting an important question.  The method of incorporating higher-level structure and structure interaction was also found to be novel and effective.  However, there were some common concerns:
1. the paper is missing key baselines (*p6sz*, *dbpi*)
2. performance is not reported for standard data splits (*p6sz*).  The model should be tested against other benchmarks to rule out bias (*dbpi*)
3. lack of ablation study on drug representations (*dbpi*)
4. need for additional results analysis (*dbpi*)
5. the technical presentation should be improved and the mathematical equations require editing (*znRg*)
6. lack of novelty (*FX9M*)

**Reviewer Concerns:**

Since there was no rebuttal, the reviewers would probably not change their scores or consider their concerns addressed.

**Reviewer Scores:**

The scores stand at: 4,4,6,2.  The majority of reviewers are in favor of rejection.

---

### Decision · Program_Chairs · 2026-01-26

Reject